# Apolipoprotein ɛ4 Status and Brain Structure 12 Months after Mild Traumatic Injury: Brain Age Prediction Using Brain Morphometry and Diffusion Tensor Imaging

**DOI:** 10.3390/jcm10030418

**Published:** 2021-01-22

**Authors:** Torgeir Hellstrøm, Nada Andelic, Ann-Marie G. de Lange, Eirik Helseth, Kristin Eiklid, Lars T. Westlye

**Affiliations:** 1Department of Physical Medicine and Rehabilitation, Oslo University Hospital, 0424 Oslo, Norway; nadand@ous-hf.no; 2Research Center for Habilitation and Rehabilitation Models and Services (CHARM), Institute of Health and Society, University of Oslo, 0318 Oslo, Norway; 3Department of Psychology, Faculty of Social Sciences, University of Oslo, 0317 Oslo, Norway; a.m.g.de.lange@psykologi.uio.no (A.-M.G.d.L.); l.t.westlye@psykologi.uio.no (L.T.W.); 4LREN, Centre for Research in Neurosciences-Department of Clinical Neurosciences, CHUV and University of Lausanne, 1011 Lausanne, Switzerland; 5Department of Psychiatry, University of Oxford, Oxford OX3 7JK, UK; 6Institute of Clinical Medicine, Faculty of Medicine, University of Oslo, 0318 Oslo, Norway; ehelseth@ous-hf.no; 7Department of Neurosurgery, Oslo University Hospital, 0424 Oslo, Norway; 8Department of Medical Genetic, Oslo University Hospital, 0424 Oslo, Norway; k.l.eiklid@gmail.com; 9NORMENT, Division of Mental Health and Addiction, Oslo University Hospital, 0424 Oslo, Norway; 10K.G Jebsen Center for Neurodevelopment Disorders, Faculty of Medicine, University of Oslo, 0318 Oslo, Norway

**Keywords:** mild traumatic brain injury, APOE, brain-age gap, MRI

## Abstract

Background: Apolipoprotein E (APOE) ɛ4 is associated with poor outcome following moderate to severe traumatic brain injury (TBI). There is a lack of studies investigating the influence of APOE ɛ4 on intracranial pathology following mild traumatic brain injury (MTBI). This study explores the association between APOE ɛ4 and MRI measures of brain age prediction, brain morphometry, and diffusion tensor imaging (DTI). Methods: Patients aged 16 to 65 with acute MTBI admitted to the trauma center were included. Multimodal MRI was performed 12 months after injury and associated with APOE ɛ4 status. Corrections for multiple comparisons were done using false discovery rate (FDR). Results: Of included patients, 123 patients had available APOE, volumetric, and DTI data of sufficient quality. There were no differences between APOE ɛ4 carriers (39%) and non-carriers in demographic and clinical data. Age prediction revealed high accuracy both for the DTI-based and the brain morphometry based model. Group comparisons revealed no significant differences in brain-age gap between ɛ4 carriers and non-carriers, and no significant differences in conventional measures of brain morphometry and volumes. Compared to non-carriers, APOE ɛ4 carriers showed lower fractional anisotropy (FA) in the hippocampal part of the cingulum bundle, which did not remain significant after FDR adjustment. Conclusion: APOE ɛ4 carriers might be vulnerable to reduced neuronal integrity in the cingulum. Larger cohort studies are warranted to replicate this finding.

## 1. Introduction

Following mild traumatic brain injury (TBI) (MTBI), most individuals recover quickly; however, the rate of recovery varies considerably. Between 6% to 64% of patients experience long-term symptoms and disability [1,2,3]. This large heterogeneity in prognosis and clinical course following MTBI suggests that factors other than injury severity are important predictors of outcome. Some of the unexplained variance may arise from genetic differences in processes involved in neural repair or neurodegenerative mechanisms. Apolipoprotein E (APOE), having three common protein isoforms (ɛ2, ɛ3, and ɛ4), is involved in neuronal repair and plasticity [4] and carriers of the ɛ4 allele are at increased risk for neurological conditions such as Alzheimer’s disease (AD) [5] and poor outcome following moderate and severe TBI [6]. While the mechanisms by which the ɛ4 allele exerts an effect is unclear, *APOE* gene expression increases following brain injury [7], and it has been shown that the ɛ4 allele is less effective at promoting neuronal repair than the ɛ2 and ɛ3 alleles [4].

In addition to its effects on outcome after TBI, the *APOE* genotype may interact with TBI to increase the risk of developing neurodegenerative disorders later in life [8]. APOE ɛ4 carriers are at increased risk for cognitive impairment and chronic traumatic encephalopathy following head injury [9,10,11]. APOE ɛ4 has been shown to promote amyloid deposition in individuals with TBI [12] and the combination of MTBI and genetic risk for AD may play a role in the degeneration of structural brain integrity [13]. These results highlight the intuitive notion that clinical and neuropathological outcomes following MTBI are determined by complex interactions between injury characteristics and genetic factors.

The majority of prior studies assessing the risk of neurodegenerative diseases following TBI have focused either on the risk imparted by a TBI of any severity or on the risk imparted by moderate or severe TBI. Two systematic reviews considering the risk of dementia following MTBI [14,15] that included literature published from 1980 through 2012 identified only four qualifying studies [16,17,18,19]. These studies were limited by a relatively small number of MTBI patients, yet two reported a significant association between MTBI and dementia [17,18]. Meanwhile, in a large, prospective, population-based study of 6645 individuals aged 55 years or older who were free of dementia at baseline, Metha et al. [16] found that MTBI was not a major risk factor for AD. Moreover, brain trauma did not appear to increase the risk of AD in ɛ4 carriers. One study by Yue et al. [20] of a group of MTBI patients found that the APOE ɛ4 allele may confer an increased risk of impairment of six-month verbal memory. These authors also found that intracranial pathology was the driver of decreased verbal memory performance at six months. Finally, it has been reported reduced cortical thickness in AD-related brain regions in MTBI patients at increased genetic risk for AD [13].

In contrast to measures of brain volume and gross morphology, diffusion tensor imaging (DTI) provides measures of white matter coherence and structure. DTI measures are highly sensitive to age [21,22] and have been used to study associations between APOE status and brain white matter in healthy individuals [23] and after TBI, suggesting greater vulnerability to white matter abnormalities in ɛ4 carriers as compared with non-carriers following close-range blast exposure [24].

A critical goal of biomedical research is to establish indicators of neuropathology after TBI to identify actionable targets for treatment and improved outcome. These biomarkers are quantifiable characteristics of biological processes related to TBI and can be used as surrogates for the disease process. Recent efforts to develop neuroimaging-derived markers for brain health include brain-age prediction, which uses machine learning to predict age based on brain characteristics [25,26,27]. An individual’s estimated brain age can be compared to their chronological age to calculate deviation from normative age trajectories, often referred to as the brain-age gap (BAG). Such deviations have been associated with a range of clinical risk factors [28,29,30] as well as neurological and neuropsychiatric diseases [31,32,33,34]. Estimated brain age in TBI patients has been shown to be higher relative to their chronological age [35]. This discrepancy increases with more time since the injury, suggesting that TBI accelerates brain aging.

Brain-age prediction based on different brain imaging measures provides a window into different mechanisms of brain aging affecting brain gray and white matter. The successful identification of neurodegenerative markers in TBI patients would have important implications for the development of interventions to prevent TBI-induced neurodegeneration and planning of health service delivery such as individualized follow-ups and short- and long-term targeting impairments and disability. There is a lack of studies investigating the influence of APOE ɛ4 on intracranial neuropathology following MTBI and no investigations have conducted brain-age predictions among MTBI patients. Hence, by performing brain-age prediction based on sensitive MRI measures of brain morphometry and DTI measures of white matter architecture and coherence, we tested for associations between APOE ɛ4 status and brain-age gap in patients with MTBI at 12 months after injury. To increase the generalizability of our study, we used an independent training set for brain-age prediction. For comparison and transparency, we also report results from tests comparing ɛ4 carriers and non-carriers with regard to more conventional measures of brain morphometry and DTI measures.

## 2. Materials and Methods

### 2.1. Ethical Statement

All study protocols were approved by the Norwegian Regional Committee for Medical Research Ethics (2010/1899) (Oslo, Norway). Additionally, all participants provided written informed consent and all methods were carried out in accordance with the relevant guidelines and regulations of the Norwegian Regional Committee for Medical Research Ethics.

### 2.2. Participants and Procedure

Patients with acute MTBI admitted to Oslo University Hospital between September 2011 and September 2013 were included in this prospective cohort study and followed up for 12 months postinjury (2012–2014). Patients aged 16 to 65 years with a recent (<24 h) history of head trauma (hospitalization with an International Classification of Disease, 10th revision (ICD-10) diagnosis codes S06.0–S06.9), resulting in loss of consciousness (LOC) of less than 30 min, posttraumatic amnesia lasting (PTA) less than 24 h, and Glasgow Coma Scale score of 13 to 15 points were included. MTBI was defined using criteria from the American Congress of Rehabilitation Medicine (ACRM) [36]. Exclusion criteria for this study included the presence of severe mental illness (schizophrenia or bipolar disorder), progressive neurologic disease, ICD-10 diagnosis of substance dependence, contraindications for magnetic resonance imaging (MRI), and a lack of Norwegian language skills. We have previously reported brain volumetric and morphometric findings based on T1-weighted MRI data and DTI data in an overlapping sample [37,38,39], but not used brain-age prediction and not in relation to APOE status. Of the total sample of 168 patients at baseline, 134 (80%) patients returned for the 12-month follow-up visit, which included a clinical assessment and multimodal MRI; of these patients, 123 had available volumetric, APOE, and DTI data of sufficient quality. Thus, the final study population was 123 patients.

### 2.3. Biospecimen and Genotyping Procedures

The variants in exon 4 in *APOE* (GenBank: NM_000041)—that is, c.388T>C and c.526C>T—were analyzed by polymerase chain reaction and Sanger sequencing in DNA extracted from peripheral leukocytes. Primers were designed using Primer3Plus (Bioinformatics, Arlington, VA, USA) [40] and the polymerase chain reaction products were purified and Sanger-sequenced using an ABI 3730xl DNA analyzer and ABI BigDye terminator cycle-sequencing kits v3.1 (Thermo Fisher Scientific, Waltham, MA, USA). Sequences were analyzed with the DNA Sequencing Analysis software program (version 5.1; Applied Biosystems, Foster City, CA, USA) and the SeqScape software program (version 2.7; Thermo Fisher Scientific).

### 2.4. MRI Processing

3-Tesla MRI scans (GE Signa HDxt; GE Medical Systems, Chicago, IL, USA) data were obtained 12 months postinjury using two different head coils (Head/Neck/Spine (HNS) and 8HRBRAIN). Briefly, the protocol included a three-dimensional fast spoiled gradient-echo T1-weighted sequence used for morphometric assessments (repetition time (TR)/echo time (TE) ms/inversion time (Ti): 7.8/2.96/450 ms, flip angle (FA): 12° and spatial resolution: 1.0 × 1.0 × 1.2 mm). Acquisition parameters were optimized for increased gray/white matter contrast. In addition, a T2-weighted sequence and a T2 susceptibility-weighted angiography sequence were performed to depict hemorrhagic or other lesions. No major scanner upgrade occurred during the study period and the cortical reconstruction and segmentation processes are described in previous articles [37,38].

For diffusion-weighted imaging, a two-dimensional spin-echo whole-brain echo-planar imaging pulse with the following parameters was used: TR, 15 s; TE, 85 ms; FA, 90°; slice thickness, 2.5 mm; field of view, 240 × 240; acquisition matrix, 128 × 128; and in-plane resolution, 1.875 × 1.875. Additionally, 30 volumes with different gradient directions (b = 1000 s/mm^2^) and two b = 0 volumes with reversed phase-encode (blip up/down) were acquired. Image analyses and extraction of atlas-based regions of interests were completed using FMRIB Software Library (FSL) [41] and tract-based spatial statistics (TBSS) [42], as previously described [39]. We report on fractional anisotropy (FA) and mean diffusivity (MD), estimated using dtifit in FSL. FA and MD measure demyelination as a sign of white matter alteration [43] and have been used as markers of structural damage in some pathologies, e.g., mild cognitive impairment and Alzheimer’s disease [44]. Results from axial diffusivity (AD) and radial diffusivity (RD) are presented in the Appendix A.

### 2.5. Brain-Age Prediction

In line with previous research [45,46], the Cambridge Centre for Ageing and Neuroscience (Cam-CAN) dataset (http://www.mrc-cbu.cam.ac.uk/datasets/camcan) was used as a training sample in the brain-age analyses. Briefly, this dataset includes information from healthy participants aged 18 to 87 years who were recruited through a research project funded by the Biotechnology and Biological Sciences Research Council, the UK Medical Research Council and University of Cambridge [47,48]. For more information, see http://www.cam-can.org. After quality control, MRI data from 622 participants were included in the training sample (age range: 18–87 years, mean age ± standard deviation: 54.17 ± 18.38 years).

The MRI features included 276 measures for DTI and 269 measures for the T1-weighted data, as described in Section 2.4. Age-prediction models were run using the XGBoost regression model, which is based on a decision-tree ensemble algorithm (https://xgboost.readthedocs.io/en/latest/python). XGboost includes advanced regularization to reduce overfitting [49] and incorporates a gradient-boosting framework where the final model is based on a collection of individual models (https://github.com/dmlc/xgboost). The parameters were set to a maximum depth of 3, number of estimators of 180, and learning rate of 0.1 based on a grid search with five folds for optimization. In addition, a grid search with nested cross-validation was performed to test for potential overfitting. The models were validated using five-fold cross-validation with 100 repetitions and the values of R^2^, root mean square error (RMSE), and mean absolute error (MAE) were calculated to evaluate model performance within the training sample.

Next, the models were applied to the sample of MTBI patients and R^2^, RMSE, and MAE values were calculated to evaluate prediction accuracy. The predicted age for each patient was derived using the Scikit Learn library (https://scikit-learn.org/stable/index.html). To adjust for a commonly observed age-bias (i.e., overestimated predictions for younger participants and underestimated predictions for older participants) [50,51], we applied a statistical correction by first fitting Y = α×Ω+ β in the Cambridge Centre for Ageing and Neuroscience dataset training sample, where Y is the modelled predicted age as a function of chronological age (Ω) and α and β represent the slope and intercept, respectively. We then used the derived values of α and β to correct the predicted age in the MTBI sample using the equation corrected predicted age = predicted age + $\left[\Omega -\left(\alpha \times \Omega +\beta \right)\right]$, before recalculating R^2^, RMSE, and MAE. BAG values for each participant were calculated using (corrected predicted age−chronological age).

### 2.6. Statistical Analyses

Descriptive statistical analyses were performed using the Statistical Package for the Social Sciences for Windows (version 25; IBM Corporation, Armonk, NY, USA). Sample characteristics are presented as the group mean with standard deviation (SD). Differences between groups concerning continuous variables were tested using the Student’s *t*-test. The chi-squared test for contingency tables was conducted to detect group differences in categorical variables.

As the primary assessment, we performed multiple linear regression analysis to test for differences between APOE ɛ4 carriers and non-carriers in BAG based on T1-weighted and DTI data, while covarying for age, sex, and head coil. Because of the number of tests performed, we controlled the false discovery rate (FDR) using the Benjamini−Hochberg procedure [52]. Briefly, all *p* values were ordered from smallest to largest. The smallest *p* value was given a rank of *i* = 1, then next smallest *i* = 2, etc. We then compared each individual *p* value to its Benjamini−Hochberg critical value, (i/m)Q, where *i* is the rank, *m* is the total number of tests, and *Q* is the FDR you choose. The largest *p* value that has *p* < (i/m)Q is significant, and all *p*-values smaller than it are also significant. We present the raw *p* values and those significant at FDR = 0.1.

As follow-up analysis, we conducted additional linear regressions to test for group differences in individual MRI features, including intracranial volume, age, sex, and head coil in the models for volume measures and age, sex, and head coil for cortical thickness and DTI.

Regression results are presented as β coefficients, standard errors of β, *p*-values, and explained variance (R^2^).

### 2.7. Demographic and Clinical Assessment

We assessed key preinjury, acute, and postinjury variables. Acute clinical data were obtained from patient medical records and MRI and APOE assessments at 12 months of follow-up. Information regarding age, sex, and education level was obtained from clinical interviews.

The Glasgow Coma Scale (GCS) [53] assesses the conscious state of the patient at injury site or/and admission to the hospital and mild severity scores range from 13 to 15 points (from alert to well-orientated). The duration of posttraumatic amnesia was assessed in the emergency department and dichotomized into yes/unknown or no. The presence and duration of LOC were confirmed based on medical records and classified into no LOC or LOC/unknown. Causes of injury were obtained from patient medical records and classified as traffic accident, fall, violence, or other. Education was measured in years. Glasgow Outcome Scale-Extended (GOSE) was used as measure of global functions including independence, work, social and leisure activities, and participation in social life [54]. It is an 8-point ordinal scale reflecting good recovery (>7), moderate (5, 6) and severe (3, 4) disability, vegetative state (2), and death (1). Rivermead post-concussion symptoms questionnaire (RPQ) was used to assess cognitive, emotional and somatic symptoms [55]. Patient Health Questionnaire 9 (PHQ-9) was used to assess depressive symptoms [56]. Total score of 0–4 indicate no depression, 5–9 mild, 10–14 moderate, 15–19 moderately severe, and 20–27 severe depression, and the total score was used.

## 3. Results

### 3.1. Comparison Groups, Demographics, and Injury-Related Variables

The study population (*n* = 123 MTBI cases that met all inclusion criteria) was 61% men, with a mean age of 39 years and, on average, 15 years of education. When stratified by APOE ɛ4 status, the ɛ4+ and the ɛ4− groups did not differ significantly in any demographic or clinical variable (Table 1). APOE genotypes were distributed as follows: 1 case of ɛ2/ɛ2 (1%), 11 cases of ɛ2/ɛ3 (9%), 63 cases of ɛ3/ɛ3 (51%), 41 cases of ɛ3/ɛ4 (33%), 3 cases of ɛ2/ɛ4 (3%), and 4 cases of ɛ4/ɛ4 (3%). Overall, 39% of patients carried at least one APOE ɛ4 allele. Due to the relatively small sample size we were unable to test for differences between homozygote and heterozygote APOE ɛ4 carriers. There were no differences between the APOE ɛ4 groups between complicated or uncomplicated MTBI or in type of injury or location.

Table 2 presents means and standard deviation (SD) of GOSE, RPQ, and PHQ-9 assessed at 12 months follow-up for the two APOE groups. Results indicate overall good functional outcome (GOSE mean 7.2 (SD 0.82)), low symptom burden (RPQ mean 13.12 (SD 13.8)), and PHQ-9 mean 6.50 (SD 5.16). There were no significant differences between the APOE ɛ4+ and the ɛ4− groups.

### 3.2. Brain-Age Prediction

The accuracies of the age prediction models are shown in Table 3. Within the training sample, the average R^2^ result presented mean ± standard deviation values of 0.82 ± 0.03 for the DTI model and 0.81 ± 0.03 for the model based on T1-weighted data. The results from the nested cross-validation approach for hyper-parameter optimization were similar to the main results, as shown in Table 3.

When the models were applied to the MTBI patients, the R^2^ values were 0.76 for the DTI model and 0.78 for the model based on T1-weighted data after age-bias correction was applied. The correlation between predicted and chronological age was *r* = 0.76 (95% confidence interval (CI): 0.68–0.83) for the DTI model and 0.74 (95% CI: 0.66–0.82) for the model based on T1-weighted data. Meanwhile, after age-bias correction, the correlations showed *r* = 0.87 (95% CI: 0.82–0.91) for the DTI model and *r* = 0.88 (95% CI: 0.84–0.92) for the model based on T1-weighted data, as shown in Figure 1.

### 3.3. Association between APOE Status and Brain-Age Gap

Table 4 summarizes the respective brain-age gaps in APOE ɛ4 carriers and non-carriers. Multiple regression analyses revealed no significant differences between the groups.

### 3.4. Association between APOE Status and Brain Morphometry

Table 5 summarizes the mean (SD) neuroanatomic volumes and cortical measures per region of interest (ROI) and Table 6 presents the mean value of cortical thickness per ROI at 12 months postinjury. Table 7 summarizes the results from the multiple regressions testing for group differences; no significant differences between groups were found.

Multiple regressions revealed no significant between-group differences in total intracranial volume, left or right hemisphere thickness or total cortical volume. There was a nominally significant (*p* = 0.02) difference between APOE ɛ4 carriers as compared with non-carriers in brainstem volume, which did not remain after FDR adjustment.

### 3.5. Association between APOE and DTI Measures

Table 8 shows the results from ROI-based analyses. Multivariable regressions revealed lower FA in APOE ɛ4 carriers as compared with non-carriers in the hippocampal part of the cingulum bundle in the right hemisphere (*p* = 0.01), but this effect did not remain significant after FDR adjustment. No other significant group differences were found.

## 4. Discussion

The clinical outcome following MTBI is characterized by substantial individual variability and, while some patients suffer from long-term difficulties and decline, others experience only mild symptoms and a swift recovery. APOE ɛ4 is associated with an increased risk of unfavorable long-term (≥6 months) functional outcome [57] after TBI. Combining advanced brain imaging and information on genetic risk for neurodegenerative disorders may provide insight into the underlying pathophysiology related to functional impairment following MTBI.

APOE ɛ4 is the strongest genetic risk factor for AD and has also been associated with a greater risk for poor outcome following TBI. To date, the existing literature on APOE and TBI mostly covers functional outcome and there is a lack of studies on APOE in association with brain-structural changes and neurodegeneration following mild TBI. In this study, we combined different MRI approaches to study the associations between mild TBI, APOE ɛ4 status, and brain gray and white matter structure. In sum, our main analysis revealed no significant differences between APOE ɛ4 carriers and non-carriers in estimated brain-age gap, and follow-up analysis confirmed no significant differences in subcortical and cortical volumes, cortical thickness, or DTI metrics.

The lack of significant group differences may have several explanations. First, MRI was performed 12 months postinjury. The volume trajectory after MTBI is not known and available brain volumetric research focusing on MTBI is limited. A one-year interval between the injury and brain scan may be too short to identify long-term brain degeneration and atrophy, which may not happen until several years later [58]. The time point at which the brain begins atrophying after TBI and whether there are different long-term longitudinal brain changes for APOE ɛ4 carriers and non-carriers beyond the current interval of 12 months remain unclear. We did not observe differences in brain structures known to be involved in AD such as reductions in cortical thickness in the medial temporal lobe or hippocampus volumes. While APOE ɛ4 is the major genetic risk factor for late-onset AD [59], our results also underscore the need for powerful genome-wide and polygenic approaches to studying complex phenotypes such as clinical outcome and brain changes following TBI. Secondly, it could be that the measurements used in this study are not sensitive enough and that other objective markers of biological aging could provide additional information. For example, reduction in telomere length, a biomarker of cellular senescence and neurological health, is implicated in aging and neurodegenerative diseases (e.g., Alzheimer’s) [60] and has been associated with mild head injuries [61]. In a study by Li et al., they investigated other assessment of biological age like DNA methylation age estimator (DNAmAge), physiological age, cognitive function, functional aging index (FAI), and frailty index (FI) and found that the largest effects of mortality risk were seen for methylation age estimators and the FI [62]. There are also other biomarkers of interest when it comes to MTBI and aging pathobiology, like inflammatory biomarkers such as interferons, cytokines, and interleukins [63,64], and biomarkers for oxidative stress [65] which should be investigated further in future studies.

We did however find a difference in the brain-stem volume between the two APOE ɛ4 groups, albeit not a significant one. There is a lack of studies assessing mild TBI in this region and no studies have explored the association with APOE. Hemorrhagic lesions and axonal degeneration, however, can be seen in the brain stem after experimental mild to moderate neurotrauma [66]. Additionally, animal autopsy studies have shown that, even after very mild head trauma, microglia clusters indicative of axonal damage are most prominent in brain stem white matter [67]. Garman et al. [68] exposed body-shielded rats to a single blast exposure to examine neuropathological characterizations of the underlying brain tissue and found that, although neuronal and synaptic terminal degeneration improved over time, the primary pathology that persisted and worsened in the long-term was axonal injury and degeneration involving axons in the brain stems. Elsewhere, Delano-Wood et al. [69] found a link between the integrity of the brain stem’s white matter and injury severity. Brain-stem plaques were found in AD as well [70]. Taken together, these findings suggesting the brain stem as a possible vulnerable region in APOE ɛ4 (+) patients should be further examined.

DTI has been used to assess microstructural features of white matter [71]. Because of their length, axons are particularly vulnerable to mechanical injury and both clinical and experimental studies have repeatedly reported that mild, moderate, and severe TBI can cause axonal injury [72,73]. Postmortem case analysis has revealed diffuse axonal injury to be the most common kind of damage apparent after MTBI [74], while neuropathological features of dementia—for example, β-amyloid burden and neurofibrillary tangles—follow persistent axonal degeneration [75]. FA is the most commonly used DTI measure and is affected by many factors, including axonal degeneration, demyelination, disorganization, packing density, and other microstructural features, and should be interpreted as an indirect proxy of white matter integrity. Although not significant after FDR adjustment, we found lower FA in the left and right cingulum hippocampi in the APOE ɛ4 (+) group. Investigations of white matter connectivity changes in aging and AD have focused on the fornix and the cingulum as the major links between the limbic system and the rest of the brain. Specifically, the cingulum connects the cingulate and the parahippocampal gyri to the septal cortex and several studies have reported white matter changes in the cingulum in mild cognitive impairment and mild AD cases [76,77,78]. In a review and meta-analysis looking into the relationship between white-matter integrity and posttraumatic cognitive deficits after TBI, the fornix and the cingulum were particularly strongly associated with the memory domain [79]. Although not significant after FDR adjustment our results may suggest that APOE ɛ4 carriers are vulnerable to reduced neuronal integrity in the cingulum, which is associated with memory and, further, with mild cognitive impairment and AD; therefore, we will investigate the association between APOE ɛ4 and cognitive function in future work. In this study, e.g., proof of concept for the neuroimaging and APOE biomarkers, we will provide in-depth analysis of association between post-traumatic symptoms, cognitive function, MRI, and APOE in a subsequent publication.

## 5. Strengths and Limitations

The strengths of this study include its prospective recruitment from consecutive admissions to a single regional trauma center and the use of multimodal brain MRI. However, the study also has several limitations. First, the study population was drawn from neurosurgical admissions and therefore, most likely, includes cases on the more severe end of the MTBI spectrum. However, no evidence of an interaction between severity and APOE ɛ4 (+/−) status was found. Secondly, this study was limited by a relatively low power in particular for testing of differences between heterozygote and homozygote APOE ɛ4 carriers. Although our selective recruitment of patients without drug abuse, mental illness, or a lack of Norwegian language skills limits the generalizability of the results, the careful exclusion of these factors and control for confounding variables allows for greater confidence in drawing conclusions. Thirdly, because of the lack of a control group either without injury or with non-head injury, we were unable to determine whether the results are specific to MTBI. The cross-sectional design is also a limitation and longitudinal studies are needed to explore the time course of brain structural changes.

## 6. Conclusions

Our analyses revealed no significant associations between brain-age gap, volumetric measures, or cortical thickness and APOE ɛ4 status. The suggestive lower FA in the hippocampal part of the cingulum in APOE ɛ4 carriers suggests that this region could be vulnerable in APOE ɛ4 carriers following MTBI and it is therefore important to pursue this finding in larger cohort studies and meta-analyses.

## Figures and Tables

**Figure 1 jcm-10-00418-f001:**
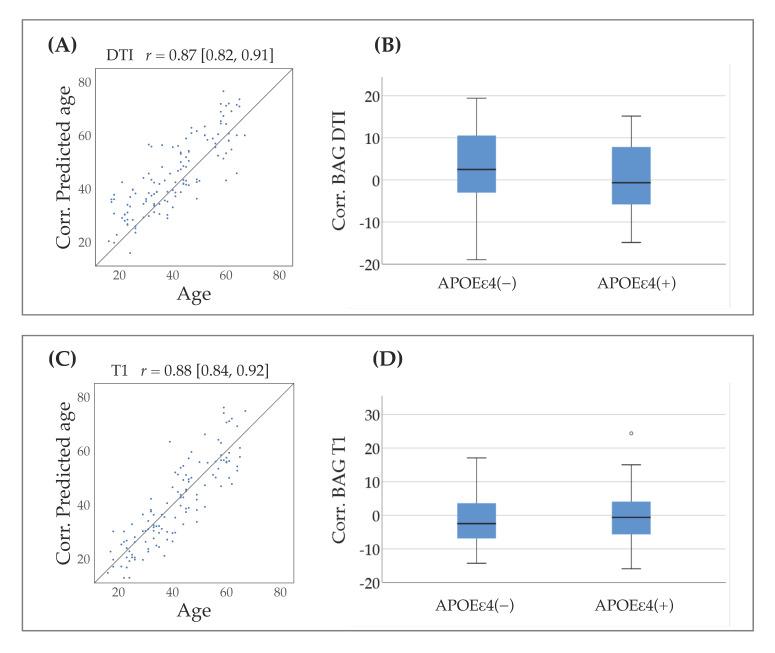
Brain-age prediction and BAG distributions per APOE group. (**A**) Chronoligcal age (x-axis) versus DTI-based corrected predicted age (y-axis; corrected for age-bias as described in Section 2.5). (**B**) The distribution of corrected DTI-based BAG estimated per APOE group (carriers versus non-carriers). (**C**) Chronological age (x-axis) versus T1-based corrected predicted age (y-axis). (**D**) The distribution of corrected T1-based BAG estimates per APOE group (carriers versus non-carriers). Abbreviations: Corr. = corrected, BAG = Brain-age gap.

**Table 1 jcm-10-00418-t001:** Demographics and injury related variables of apolipoprotein E (APOE) ɛ4 +/−mild traumatic brain injuries (MTBIs) *n* = 123.

Variables	Overall	APOEɛ4(−)	APOEɛ4(+)	*p*-Value
	(*n* = 123)	(*n* = 75)	(*n* = 48)	
Age years, mean (SD)	39.3 (14.0)	40.7 (14.2)	37.3 (13.6)	0.19
Gender (*n*, %)				
- male	75 (61)	45 (60)	30 (62)	0.78
- female	48 (39)	30 (40)	18 (38)	
Education (years)	14.7 (2.8)	14.9 (2.5)	14.3 (3.3)	0.33
Mechanism of injury (*n*, %)				
- Traffic accidents	52 (42)	31 (41)	21(43)	0.63
- Falls	46 (37)	31 (41)	15 (31)	
- Violence	13 (11)	7 (10)	6 (13)	
- Other	12 (10)	6 (8)	6 (13)	
GCS				
13	6 (5)	4 (5)	2 (4)	0.55
14	29 (24)	20(27)	9 (19)	
15	88 (71)	51(68)	37(77)	
LOC (*n*, %)				
- no	25 (20)	18 (24)	7 (15)	0.21
- yes/unknown	98 (80)	57 (76)	41(85)	
PTA (*n*, %)				
- no amnesia	12 (10)	9 (12)	3 (6)	0.29
- yes/unknown	111 (90)	66 (88)	45(94)	
Complicated				
-no	67 (54)	41 (55)	26 (54)	0.96
- yes	56 (46)	34 (45)	22 (46)	
Coil (*n*, %)				
-HNS	37 (30)	20 (27)	17 (35)	0.3
-8HRBRAIN	86 (70)	55 (73)	31 (65)	

GCS: Glasgow Coma Scale; LOC: loss of consciousness; PTA: Posttraumatic amnesia.: *p*-values: *T*-test for continuous variables; Chi square for categorical variables.

**Table 2 jcm-10-00418-t002:** Self-reported outcome measures of patients on APOE ɛ4 status.

Variables	Overall (*n* = 123)	APOEɛ4(−) (*n* = 75)	APOEɛ4 (+) (*n* = 48)	*p*-Value
RPQ total				
Mean (SD)	13.12 (13.81)	12.23 (13.54)	14.52 (14.26)	0.38
GOSE				
Mean (SD)	7.20 (.82)	7.21 (.84)	7.19 (.79)	0.86
PHQ 9				
Mean (SD)	6.50 (5.16)	6.23 (4.83)	6.94 (5.66)	0.47

RPQ: Rivermead post-concussion symptoms questionnaire; GOSE: Glasgow Outcome Scale-Extended; PHQ-9: Patient Health Questionnaire 9; *p*-values: *t*-test for continuous variables. SD: standard deviation.

**Table 3 jcm-10-00418-t003:** Brain age prediction.

**Model Performance—Training Sample (Cam-CAN)**						
Model	R^2^	RMSE	MAE	R^2^_nestedCV_	RMSE_nestedCV_	MAE_nestedCV_
DTI	0.82 ± 0.03	7.81 ± 0.48	6.17 ± 0.40	0.82 ± 0.03	7.75 ± 0.62	6.13 ± 0.52
T1	0.81 ± 0.03	7.97 ± 0.50	6.26 ± 0.42	0.81 ± 0.02	8.07 ± 0.29	6.39 ± 0.21
**Model Performance—Test Sample (MTBI)**						
Model	R^2^	RMSE	MAE	R^2^ _corr_	RMSE _corr_	MAE _corr_
DTI	0.58	11.89	9.76	0.76	9.01	7.48
T1	0.56	10.35	8.63	0.78	7.65	6.20

R^2^, root mean square error (RMSE), and mean absolute error (MAE) for the age prediction models within the training sample (the Cambridge Centre for Ageing and Neuroscience, Cam-CAN), and when applied to the test sample (MTBI patients). For the model validation within the training sample, means and standard deviations are provided based on 5-fold cross validations with 100 repetitions. *NestedCV* indicates the values based on nested cross-validation for hyperparameter optimization. For the test sample, R^2^, RMSE, and MAE are provided before and after age-bias correction (described in Section 2.5).

**Table 4 jcm-10-00418-t004:** Multivariable regression of APOE ɛ4status and T1 and DTI brain age gap.

Brain Age Gap	Comparison Groups			
	APOE-ɛ4(−) vs. APOE-ɛ4(+)			
	B	SE	*p*-Value	R^2^
T1w-based:				0.054
-APOE ɛ4	0.790	1.404	0.58	
-Age (per year)	0.032	0.050	0.52	
-Sex	1.791	1.424	0.21	
-Head coil	−3.328	1.508	0.03	
DTI-based:				0.119
-APOE ɛ4	−2.564	1.563	0.1	
-Age (per year)	0.131	0.056	0.02	
-Sex	−3.614	1.586	0.02	
-Head Coil	−3.425	1.679	0.04	

**Table 5 jcm-10-00418-t005:** APOE ɛ4 status on mean values of neuroanatomic volume measurements per region of interest (ROI).

ROI	APOE-ɛ4(−) (*n* = 75)		APOE-ɛ4(+) (*n* = 48)		*p*-Value
	Mean mm^3^	SD mm^3^	Mean mm^3^	SD mm^3^	
Total ICV	1.584767	152.613	1.571603	150.490	0.64
Total gray volume	662.282	66.495	664.349	64.095	0.87
Cortex volume	491.343	51.833	492.455	47.444	0.91
L Accumbens area	645	137	688	135	0.09
R Accumbens area	649	125	636	131	0.59
L Amygdala	1597	233	1607	195	0.81
R Amygdala	1843	249	1852	240	0.84
Brainstem	21,267	2108	20.485	2514	0.06
L Caudate	3815	573	3954	517	0.18
R Caudate	3999	629	4195	568	0.08
CC-posterior	990	147	982	168	0.78
CC-mid-posterior	432	91	402	89	0.08
CC-central	437	78	414	84	0.13
CC-mid-anterior	456	94	442	79	0.39
CC-anterior	905	144	890	141	0.58
L Hippocampus	4555	542	4480	478	0.44
R Hippocampus	4551	467	4576	446	0.78
L Pallidum	1337	296	1308	312	0.61
R Pallidum	1596	280	1558	291	0.47
L Putamen	60,441	892	6009	1009	0.84
R Putamen	5946	796	6009	856	0.68
L Thalamus	8612	1183	8603	1068	0.97
R Thalamus	6983	913	6975	900	0.96
L lateral Ventricle	9760	5507	10.717	6399	0.38
R lateral Ventricle	9068	6041	8983	4994	0.93

Abbreviations: ICV = intracranial volume, L = left, R = right, CC = corpus callosum, *p*-values are from the mean comparisons by the independent *T*-tests.

**Table 6 jcm-10-00418-t006:** APOE ɛ4 status on the mean values of neuroanatomic thickness measurements per ROI.

ROI	APOE-ɛ4(−) (*n* = 75)		APOE-ɛ4(+) (*n* = 48)		*p*-Value
	Mean mm	SD mm	Mean mm	SD mm	
L hemisphere, mean	2.51	0.127	2.52	0.105	0.46
R hemisphere, mean	2.48	0.118	2.49	0.099	0.50
L frontal	2.51	0.145	2.51	0.126	0.95
R frontal	2.43	0.124	2.44	0.115	0.85
L temporal	2.95	0.151	2.97	0.136	0.45
R temporal	2.89	0.149	2.90	0.131	0.48
L parietal	2.29	0.132	2.30	0.139	0.67
R parietal	2.30	0.146	2.31	0.121	0.67
L cingulate	2.58	0.175	2.62	0.201	0.21
R cingulate	2.61	0.196	2.59	0.167	0.59
L occipital	2.04	0.111	2.05	0.100	0.55
R occipital	2.06	0.120	2.07	0.100	0.64
L insula	3.08	0.175	3.10	0.130	0.62
R insula	3.03	0.171	3.04	0.166	0.72

Abbreviations: L = left, R = right, *p*-values are from the mean comparisons by the independent *T*-tests.

**Table 7 jcm-10-00418-t007:** Multivariable regression of APOE-ɛ4 status and neuroanatomic volume and thickness per ROI.

ROI	Comparison Groups			
	APOE-ɛ4(−) vs. APOE-ɛ44(+)			
	B	SE	*p*-Value	R^2^
Total ICV	−13698	23822	0.57	0.312
Total gray volume	−1932.9	5208.5	0.71	0.825
Cortex volume	−1125.3	4541.6	0.81	0.773
L Accumbens area	22.527	19.780	0.26	0.428
R Accumbens area	−24.698	19.020	0.20	0.382
L Amygdala	−6.604	33.963	0.85	0.334
R Amygdala	12.443	42.132	0.77	0.185
Brain Stem	−710.59	306.40	0.02	0.512
L Caudate	95.763	77.147	0.22	0.468
R Caudate	135.66	83.163	0.11	0.492
L Hippocampus	−91.780	74.287	0.22	0.434
R Hippocampus	9.805	67.792	0.89	0.399
L Putamen	−143.81	124.25	0.25	0.516
R Putamen	−53.235	97.863	0.59	0.606
Left Thalamus	34.182	170.03	0.84	0.384
Right Thalamus	−25.139	125.03	0.84	0.475
CC-posterior	5.094	25.748	0.84	0.239
CC-mid-poster	−30.256	16.590	0.07	0.089
CC-central	−22.641	14.790	0.13	0.084
CC-mid-anterior	−12.932	15.477	0.41	0.157
CC-anterior	−7.022	24.002	0.77	0.216
L Pallidum	−56.903	47.265	0.24	0.317
R Pallidum	−6.0494	41.844	0.15	0.402
L lateral Ventricle	1669.5	945.24	0.08	0.287
R lateral Ventricle	555.08	933.86	0.55	0.246
L hemisphere, mean thick	0.006	0.019	0.75	0.304
R hemisphere, mean thick	0.005	0.018	0.80	0.279
L frontal thick	−0.007	0.020	0.73	0.395
R frontal thick	0.001	0.019	0.95	0.266
L temporal thick	0.011	0.024	0.63	0.257
R temporal thick	0.008	0.024	0.73	0.235
L parietal thick	−0.005	0.022	0.81	0.241
R parietal thick	−0.001	0.023	0.96	0.208
L cingulate thick	0.024	0.027	0.37	0.424
R cingulate thick	−0.029	0.028	0.30	0.351
L occipital thick	0.004	0.019	0.82	0.129
R occipital thick	0.002	0.020	0.94	0.129
L insula thick	−0.004	0.026	0.89	0.266
R insula thick	−0.007	0.028	0.81	0.231

Adjusted for total intracranial volume, sex, age, and head coil for volumes and adjusted for sex, age, and head coil for thickness. Abbreviations: L = left, R = right, CC = corpus callosum, Thick = thickness.

**Table 8 jcm-10-00418-t008:** Multivariable regression of APOE-ɛ4 status and DTI (fractional anisotropy (FA) and mean diffusivity (MD)).

ROI	Comparison Groups			
	APOE-ɛ4(−) vs. APOE-ɛ4(+)			
	B	SE	*p*-Value	R^2^
FA-ATR L	0.003	0.003	0.38	0.257
FA-ATR R	0.003	0.003	0.32	0.253
FA-CG L	0.001	0.004	0.87	0.317
FA-CG R	0.000	0.004	0.93	0.200
FA-CING L	−0.011	0.005	0.03	0.214
FA-CING R	−0.011	0.004	0.01	0.222
FA-CST L	0.001	0.003	0.83	0.253
FA-CST R	−0.002	0.003	0.61	0.289
FA-FMAJ	0.002	0.003	0.59	0.244
FA-FMIN	0.004	0.004	0.26	0.430
FA-IFOF L	0.001	0.003	0.68	0.382
FA-IFOF R	0.004	0.003	0.24	0.315
FA-ILF L	−0.001	0.003	0.79	0.348
FA-ILF R	−0.001	0.003	0.86	0.274
FA-SLF L	0.002	0.003	0.44	0.296
FA-SLF R	0.002	0.003	0.41	0.328
FA-SLFT L	0.000	0.004	0.95	0.148
FA-SLFT R	0.002	0.003	0.60	0.277
FA-UF L	0.002	0.003	0.54	0.209
FA-UF R	0.004	0.003	0.18	0.206
FA-CCBody	−0.005	0.006	0.41	0.249
FA-CCGenu	0.004	0.005	0.41	0.346
FA-CCSplenium	0.002	0.004	0.64	0.196
FA-ws	0.001	0.003	0.68	0.351
MD-ATR L	7.319 × 10^−6^	0.000	0.24	0.133
MD-ATR R	3.674 × 10^−6^	0.000	0.62	0.100
MD-CG L	8.128 × 10^−8^	0.000	0.99	0.033
MD-CG R	−1.644 × 10^−6^	0.000	0.72	0.039
MD-CING L	1.377 × 10^−5^	0.000	0.08	0.215
MD-CING R	7.395 × 10^−6^	0.000	0.38	0.117
MD-CST L	4.134 × 10^−6^	0.000	0.36	0.132
MD-CST R	5.745 × 10^−6^	0.000	0.25	0.151
MD-FMAJ	1.775 × 10^−7^	0.000	0.97	0.362
MD-FMIN	−2.931 × 10^−6^	0.000	0.66	0.191
MD-IFOF L	−2.129 × 10^−6^	0.000	0.66	0.063
MD-IFOF R	−5.288 × 10^−6^	0.000	0.29	0.100
MD-ILF L	−2.673 × 10^−6^	0.000	0.60	0.060
MD-ILF R	−5.404 × 10^−6^	0.000	0.29	0.066
MD-SLF L	−7.339 × 10^−7^	0.000	0.86	0.029
MD-SLF R	−1.968 × 10^−6^	0.000	0.65	0.051
MD-SLFT L	−2.722 × 10^−6^	0.000	0.58	0.042
MD-SLFT R	−5.847 × 10^−6^	0.000	0.29	0.154
MD-UF L	−4.273 × 10^−6^	0.000	0.43	0.266
MD-UF R	−2.661 × 10^−6^	0.000	0.64	0.207
MD-CCBody	5.241 × 10^−6^	0.000	0.46	0.100
MD-CCGenu	−2.526 × 10^−8^	0.000	0.99	0.159
MD-CCSplenium	2.404 × 10^−6^	0.000	0.70	0.213
MD-ws	2.734 × 10^−7^	0.000	0.95	0.041

Adjusted for age, sex, and head coil. Abbreviations: fractional anisotropy (FA), mean diffusivity (MD), anterior thalamic radiation (ATR), cingulum (cingulate gyrus, CG), cingulum (hippocampus, CING), corticospinal tract (CST), forceps major (FMAJ), forceps minor (FMIN), inferior fronto-occipital fasciculus (IFOF), inferior longitudinal fasciculus (ILF), superior longitudinal fasciculus (SLF), superior longitudinal fasciculus (temporal part, SLFT), uncinate fasciculus (UF), corpus callosum (CC), whole skeleton (ws).

## Data Availability

The data in this study are available on request from the corresponding author. The data are not publicly available due to data privacy and ethical restrictions.

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
