# Peer review of "Apolipoprotein ɛ4 Status and Brain Structure 12 Months after Mild Traumatic Injury: Brain Age Prediction Using Brain Morphometry and Diffusion Tensor Imaging"

_jcm, 2021, doi:10.3390/jcm10030418_

Round 1
Reviewer 1 Report
Comments on: “Apolipoprotein 4 status and brain structure 12 months after mild traumatic brain injury: brain age prediction using brain morphometry and diffusion tensor imaging”
In this study, the authors investigated the consequence of carrying APOE e4 alleles on long-term outcome after mild traumatic brain injury (MTBI) in a cohort of 123 patients. To evaluate the outcome, the authors chose up-to-date magnetic resonance imaging (MRI) techniques. More specifically, T1- and tensor diffusion sequences were used to measure the brain morphometry and the brain-age gap (BAG), which correspond to a prediction of the brain age in MTBI patients (APOE e4 carriers and non-carriers) compared to a control non-injured patient database. Even if the authors already used this cohort in other studies, they focused here on APOE e4 genotype as well as BAG which consist of the novelties of the current study. The overall message of the study is that there is no obvious change on long term outcome after MTBI in patients that carry the APOE e4 alleles. Some results showed some trend to a worse outcome, such as brainstem volume and lower diffusion index in the cingulum hippocampi, but failing statistical significance. Overall, this study is scientifically relevant for the field of TBI but needs to be rearranged to be more reader friendly, and by highlighting only the main results (lots of data do not serve the current study and should be implemented as supplementary material). The literature cited should also be revised (1/7 of references from the 1990’s, early 2000’).
More detailed comments:
- Why did the authors focus on 12 months after MTBI? Do the authors have data concerning the 4 weeks post MTBI? And what about data concerning APOE e4 carriers’ patients?
- Did the authors look at the brain morphometry/BAG by comparing control and MTBI patients? Also, is there any dataset available to compare control (non-injured) vs APOE e4 carriers’ patients? Maybe carrying the alleles has already some effect somehow on brain morphometry/BAG.
- The effect shown on the brainstem is concerning. Do the authors can provide data concerning the site of injury? One possible cause of brainstem changes could be due to the contrecoup injury.
- Were there any patients included with an apparent lesion on MRI? Probably not, but it is always important to mention it.
- In the spirit of being more reader friendly, the authors should provide MRI images of T1 and DTI, and maybe represent their more important results by using box plots, and transfer the other data in supplementary material, if possible.
- Why the authors did not measure radial diffusivity (RD) and axial diffusivity (AD)?
- Do they authors have any data concerning amyloid deposition on these patients?
- Did the authors find correlations within noncarriers’ MTBI patients between brain morphometry/BAG and long-term cognitive impairments? Same in APOE e4 carriers?
Minor comments:
- Ref 54 is not in the good format
- Line 146: 1.875 x* 1.875, please correct.
Reviewer 2 Report
General Comments
The researchers propose to study possible changes in brain anatomy in a cohort of patients with MTBI who presented to hospital and were studied 12 months after discharge to assess possible correlations between APOE 4 status and changes in brain integrity and possible permanent anatomical changes using a score called the brain age gap, a technique with a well described history of use in AD (Franke K, Gaser C. Ten Years of BrainAGE as a Neuroimaging Biomarker of Brain Aging: What Insights Have We Gained? Front Neurol. 2019 Aug 14;10:789. doi: 10.3389/fneur.2019.00789. PMID: 31474922; PMCID: PMC6702897). The authors present the methods in an organized fashion and results are presented in tables and text that are easy to understand. The conclusions were presented clearly, and the strengths and weakness were identified in the text. No major language issues were identified, and all permissions were appropriate for the study. The conclusions included a possible change in cingulum.
Specific comments
The reader admits to a lack of sophistication regarding the statistical methods, but the authors make this readable paper easy to digest and learn from.
Introduction: no specific comments
Materials and Methods
P3 l 113 ff: The authors do not give ANY detail about clinical symptoms or sequala in the cohort. Recognizing that this is a preliminary study, e.g., proof of concept for the biomarker, I would not expect great details but there are certainly concerns that the cohort could not be segregated by symptoms either type or intensity. How this would change the outcomes is not known, but to this reader, the lack of clinical consequences of injury might help understand why differences were not found, i.e., a relatively asymptomatic cohort would be less likely to have changes at the mean or median. This represents a moderate weakness but is not necessarily reparable without a complete rewrite. But the authors could add material or text to discuss the omission.
P3 l122/p5 l205 - 206: the authors state that “a clinical assessment “was done but symptoms are not listed or reported in either general or granular ways. Most important to this reader was whether the presence of cognitive, visual, or vestibular symptoms were present (especially with the suggestion of some brainstem changes which was an important paragraph in the discussion); and of course, post traumatic headache and the phenotype expressed by groups in the cohort. As above, this is not within the Methods they proposed, but would offer the reader an opportunity to understand that the lack of difference between the E4 containing group and the null group has clinical relevance and might influence outcomes. But the authors could add material or text to discuss the omission.
P 3 l 121: The authors report 134 who returned for the 12 months follow up but do not report the original cohort (n). this may bias the sample towards those who are more engaged and may have differences in sociodemographic characteristics including self-care.
P 12 l350 – 351: The authors are careful to explain the low power and inability to differentiate homozygotes from heterozygotes, a fact that was apparent from the early part of the reading.
Results: no specific comments
Discussion and Conclusion: well written, informative and no specific comments to make
Reviewer 3 Report
This is an interesting study. Although there are limitations with the study design (some of which are already accounted for in the discussion), I believe it is worthy of publication if the below issues are addressed.
- In the abstract the authors do not define MTBI prior to first use.
- The inclusion of a control group with no mTBI would have been informative. This should be discussed in the limitation section. A longitudinal design would also be advantageous and should be acknowledged.
- Have the authors considered investigating telomere length as an additional marker of aging? A recent paper found decreased telomere length in individuals with a history of mild brain injury (Symons et al., 2020, Journal of Concussion, https://doi.org/10.1177/2059700220975609). Although it may be possible to assess telomere length with the samples used for the APOE analysis, I’m not suggesting it is vital to analyse this for the current study. However a brief discussion/acknowledgement on the topic may be of interest to the reader.
- On a related note, other biomarkers relevant to mTBI and/or aging pathobiology are of interest. Other studies have found evidence of increased inflammation, oxidative stress, vascular damage, etc (PMIDs: 32164571, 33117257, 33308001, 27458972) in individuals with a history with mTBI and it would be of interest how APOE may modify this. Again it isn’t necessary for the current study, but would be an interesting discussion point in terms of future avenues.
Round 2
Reviewer 1 Report
The authors positively addressed most of the comments from this reviewer. Few minor typos are still present though, and should be corrected before publication.
Lines 165-line 166, please reformulate "FA and MD make it possible to measure..."
Line 255: please change god to good
Line 303: please change Brainage gap to brain-age gap
Line 422: please change AOPE to APOE
Line 449: please change RAdial to Radial
Author Response
Dear reviewer
Thank you for the comments concerning our manuscript
Line 165-166, please reformulate "FA and MD make it possbile to measure..."
Response: Changed to: "FA and MD measure..."
Line 255: please change god to good.
Response: Changed god to good.
Line 422: please change AOPE to APOE.
Response: Changed AOPE to APOE
Line 449: please change RAdial to Radial.
Changed RAdial to Radial